# Antiproliferative *S*-Trityl-l-Cysteine -Derived Compounds as SIRT2 Inhibitors: Repurposing and Solubility Enhancement

**DOI:** 10.3390/molecules24183295

**Published:** 2019-09-10

**Authors:** Mohamed O. Radwan, Halil I. Ciftci, Taha F. S. Ali, Doha E. Ellakwa, Ryoko Koga, Hiroshi Tateishi, Akiko Nakata, Akihiro Ito, Minoru Yoshida, Yoshinari Okamoto, Mikako Fujita, Masami Otsuka

**Affiliations:** 1Department of Drug Discovery, Science Farm Ltd., 1-7-30-805 Kuhonji, Chuo-Ku, Kumamoto 8620976, Japan; 2Medicinal and Biological Chemistry Science Farm Joint Research Laboratory, Faculty of Life Sciences, Kumamoto University, 5-1 Oe-honmachi, Chuo-Ku, Kumamoto 8620973, Japan; 3Chemistry of Natural Compounds Department, Pharmaceutical and Drug Industries Research Division, National Research Centre, Dokki, Cairo 12622, Egypt; 4Medicinal Chemistry Department, Faculty of Pharmacy, Minia University, Minia 61519, Egypt; 5Department of Biochemistry Science, Faculty of Pharmacy, Al-Azhar University (Girls), Nasr City, Cairo 11651, Egypt; 6Seed Compounds Exploratory Unit for Drug Discovery Platform, RIKEN Center for Sustainable Resource Science, 2-1 Hirosawa, Wako, Saitama 3510198, Japan; 7Chemical Genomics Research Group, RIKEN Center for Sustainable Resource Science, 2-1 Hirosawa, Wako, Saitama 3510198, Japan; 8School of Life Sciences, Tokyo University of Pharmacy and Life Sciences, 1432-1 Horinouchi, Hachioji, Tokyo 1920392, Japan; 9Department of Biotechnology, Graduate School of Agricultural and Life Sciences, The University of Tokyo, 1-1-1 Yayoi, Bunkyo-ku, Tokyo 1138657, Japan

**Keywords:** STLC, SIRT2, anticancer, molecular docking, solubility

## Abstract

*S*-trityl-l-cysteine (**STLC**) is a well-recognized lead compound known for its anticancer activity owing to its potent inhibitory effect on human mitotic kinesin Eg5. **STLC** contains two free terminal amino and carboxyl groups that play pivotal roles in binding to the Eg5 pocket. On the other hand, such a zwitterion structure complicates the clinical development of **STLC** because of the solubility issues. Masking either of these radicals reduces or abolishes **STLC** activity against Eg5. We recently identified and characterized a new class of nicotinamide adenine dinucleotide-dependent deacetylase isoform 2 of sirtuin protein (SIRT2) inhibitors that can be utilized as cytotoxic agents based on an *S*-trityl-l-histidine scaffold. Herein, we propose new **STLC**-derived compounds that possess pronounced SIRT2 inhibition effects. These derivatives contain modified amino and carboxyl groups, which conferred **STLC** with **SIRT2** bioactivity, representing an explicit repurposing approach. Compounds **STC4** and **STC11** exhibited half maximal inhibitory concentration values of 10.8 ± 1.9 and 9.5 ± 1.2 μM, respectively, against SIRT2. Additionally, introduction of the derivatizations in this study addressed the solubility limitations of free **STLC**, presumably due to interruption of the zwitterion structure. Therefore, we could obtain drug-like **STLC** derivatives that work by a new mechanism of action. The new derivatives were designed, synthesized, and their structure was confirmed using different spectroscopic approaches. In vitro and cellular bioassays with various cancer cell lines and in silico molecular docking and solubility calculations of the synthesized compounds demonstrated that they warrant attention for further refinement of their bioactivity.

## 1. Introduction

*S*-trityl-l-cysteine **(STLC)** (Figure 1) has been identified as an ATP-noncompetitive and reversible inhibitor of human mitotic kinesin Eg5 with potential as an antimitotic chemotherapeutic agent [1,2,3,4]. **STLC** has also been reported as a potent anticancer agent in an NCI 60 tumor cell line screen (half maximal tumor growth inhibition concentration of 1.3 µM). It was listed as one of 171 molecules with a “particularly high level of interest at the NCI” in the NCI database of standard agents [5]. **STLC** development has been hindered by pharmacokinetic issues, whereas other kinesin Eg5 inhibitors from different chemical classes have already entered clinical trials. The quinazolinone derivative ispinesib was the first Eg5 inhibitor to enter phase I and phase II clinical trials and to be tested for its cytotoxic activity in patients with various tumors [6]. This was followed by more potent clinical candidates such as AZD4877, litronesib, and EMD544085 [7,8,9]. The amphiphilic character of **STLC** results in poor water solubility and reduced permeability that affect its bioavailability [10]. Unfortunately, addressing this issue by alkylation or acylation of the **STLC** free primary amine resulted in loss of activity [4]. There are many known inhibitors of Eg5 that lack a primary amine group and still have good affinity, whereas this seems to be a prerequisite for derivatives of **STLC [4,11]**. Moreover, the absence of the carboxyl group leads to an approximately 30% reduction in activity compared with the parent compound [11,12]. Modification of the carboxylic acid terminal to a primary amide or a methyl ester also reduced Eg5 ATPase activity, albeit it with reduced cellular toxicity, presumably by enhancing **STLC** cell permeation^4^. The co-crystal structure revealed polar interactions formed between the amine group of **STLC** with Glu116 and Gly117 and between its carboxyl group and Arg221 (Figure 1) [13,14]. Free ionized primary amine and carboxyl groups give the highest contribution to the binding of charged amino acids at physiological pH [15]. As a rationale, modification of the **STLC** primary amine and carboxyl groups for better pharmacokinetics may necessitate repurposing towards another valid target because such modifications may compromise its Eg5 ATPase inhibition effect.

Nicotinamide adenine dinucleotide-dependent deacetylase isoform 2 of sirtuin protein (SIRT2) is a member of the sirtuins family, which deacetylates lysine residues on histones as well as key transcriptional factors, such as p53 and NF-κB [16,17,18], and cytoskeletal proteins α-tubulin and cortactin [19,20]. It participates in the modulation of multiple biological processes including cell cycle control, genomic integrity, microtubule dynamics, cell differentiation, DNA repair, autophagy, and pathological processes such as tumorigenesis, neurodegeneration, survival, and drug resistance of cancer cells [21,22,23]. SIRT2 inhibitors showed antiproliferative effects against different cancer cell lines such as luminal and triple-negative breast cancers [22,24], leukemia of different genotypes [24,25,26,27], and cervical cancer [28]. Recently, we have identified and characterized a new class of potent and selective SIRT2 inhibitors; the lead compound, **TH-3**, has a half maximal inhibitory concentration (IC_50_) of 1.3 μM [29]. With its trityl histidine scaffold, **TH-3** shares features in common with the reported SIRT2 inhibitor SirReal2 (IC_50_ = 0.21 μM) [30]. Based on our previous findings, we sought to design **STLC**-like compounds incorporating different substitutions on the primary amine group that may imitate the trityl histidine structure. This study aimed to confer **STLC** with novel bioactivities against SIRT2 by enhancing its pharmacokinetic profile via modification of its terminal amine and carboxyl groups.

## 2. Results and Discussion

The designs of the new **STLC** derivatives were based on **TH-3** by conserving its trityl group, whereas the l-histidine moiety was replaced by l-cysteine. To obtain exhaustive structure–activity relationship (SAR) information, the amino group was reductively alkylated or acylated using different substituted aldehydes and acyl chlorides, respectively. The terminal methyl ester group was kept or converted to hydrazide (Figure 2). The synthetic procedures were initiated by **STLC** esterification using thionyl chloride and methanol to obtain compound **1** at a quantitative yield. Reductive alkylation of **1** was achieved using different commercial aldehydes in the presence of sodium triacetoxy borohydride, affording compounds **STC1–9** with moderate yields (Scheme 1). Two noncommercially available aldehydes, 4-(dimethylamino)picolinaldehyde and 4-(diethylamino)picolinaldehyde, were prepared as described previously [31,32,33]. Thereafter, the in vitro SIRT2 inhibitory activities of these synthesized compounds were examined using electrophoretic mobility shift assays [29] (Table 1).

The parent compound 1 (**STLC** methyl ester) was inactive against SIRT2. Compound STC4, with an *N,N*-dimethyl aminopyridyl moiety, demonstrated strong SIRT2 inhibition in vitro (IC_50_ = 10.8 ± 1.9 μM). Its close analogue STC9 exhibited a higher IC_50_ (17.2 ± 1.2 μM). It was notable that STC8, which lacked a dimethylamino substitution, was devoid of SIRT2 inhibition activity. Overall, 4-dimethylamino- or 4-diethylamino- substitutions in addition to the nitrogen atom of the pyridine ring seemed to be crucial for activity, which was high in accordance with previous **TH-3** SAR studies^29^.

The promising SIRT2 inhibition activity of **STC4** encouraged us to explore its SAR through further modifications. Compound **STC10** with restricted rotation of the carboxamide group was prepared by reacting synthons **3** and **1** in the presence of triethylamine (Scheme 2). Compound **3** was obtained from 4-chloropicolinic acid in two steps as reported previously [34,35]. Of note, **STC10** had no activity, indicating the importance of the free rotating C–N bond of **STC4**. To investigate the possible role of the methyl ester group of **STC4**, we converted it into its hydrazide analogue (compound **STC11**) by reaction with hydrazine hydrate in ethanol (Scheme 3). The in vitro activity was slightly improved in the hydrazide analogue to reach a single-digit micromolar level (9.5 ± 1.2 μM). The l-histidine nucleus was clearly associated with the superior activity of **TH-3** compared with **STC4**.

Next, the three active compounds (**STC4**, **STC9**, and **STC11**) were utilized for a cell-based assay to explore their potential antiproliferative effect against different cancer cell lines, including MCF7 (breast cancer), HeLa (cervical cancer), and different genotypes of leukemic cells (K562, MT-2, and HL-60) using an MTT assay. The results were compared with the gold standard **SirReal2** and **TH-3** (Table 2). Compounds **STC4** and **STC9** showed pronounced and closely similar cytotoxicity against the tested cancer cells in the low micromolar range, whereas HL-60 was exclusively more sensitive to **STC4** (IC_50_ 450 nM). Of note, compounds **STC4** and **STC9** outperformed cellular **SirReal2** activity, but we cannot rule out the presence of other off-target effects. In general, **STC11** cellular activity was less than **STC4** and **STC9**. This may be explained by the lower cellular permeability due to the presence of the more polar hydrazide group.

We then tested DNA cleavage as a possible molecular mechanism for cytotoxicity of the tested compounds using plasmid DNA (pUC19 DNA) witH- and without iron (II), H_2_O_2_, and ascorbic acid complex (Figure 3). The DNA cleavage reaction was carried out by incubating the reaction mixture at 37 °C for 2 h, and electrophoresis was performed. After electrophoresis, the DNA was stained with ethidium bromide, and the bands were visualized by exposure to ultraviolet radiation and recorded using an electronic camera. **STC9** showed the strongest DNA cleavage effect at 1 μM concentration, followed by **STC11** and **STC4**, respectively, and it was similar to **TH-3**. The results indicated that **STC4**, **STC9**, and **STC11** may generate activated oxygen and cleave DNA at nontoxic concentrations. Furthermore, these compounds may activate oxygen in the cytoplasm. This may explain their high cellular cytotoxic activity owing to a dual effect combining SIRT2 inhibition and DNA cleavage.

Next, we performed an in silico molecular docking study to explore the potential binding mode of our compounds on the SIRT2 crystal structure. **STC4** is considered as the parent of the less active derivative **STC9** and the more active derivative **STC11**. Subsequently, **STC4** was selected for our docking study to elucidate the structural mechanism of SIRT2 inhibition by this new class of compounds.

The co-crystallized ligand **SirReal2** was firstly re-docked in its corresponding co-crystal structure (PDB code 4RMG) to test whether the molecular operating environment (MOE) was able to accurately reproduce the correct binding mode of the inhibitor. We kept the conserved structural bridging water molecule (W540) because of its critical importance for the ligand binding affinity to SIRT2 active site as experimentally confirmed in a previous SAR study of potent SIRT2 inhibitors [36]. Figure 4A shows a superimposition of the co-crystallized ligand **SirReal2** and its superposed docking conformation, where **SirReal2** is perfectly docked into its crystal structure with a root-mean-squared deviation value 0.16 Å, and it formed a hydrogen bond with the conserved water molecule (W540) similar to the co-crystallized ligand.

As demonstrated in Figure 4B, the top-scoring position for **STC4** fitted into the SIRT2 active site nearly at a similar position of co-crystallized ligand **SirReal2**. Compared with the more potent SIRT2 inhibitor **TH-3 (**Figure 4C**)**, it showed that **STC4** shared similar interactions and a similar binding orientation, except the binding to the selectivity pocket. **STC4** only partially occupied the selectivity pocket, because of its smaller cysteine moiety, compared with the longer histidine moiety of **TH-3**. This shorter molecular distance of **STC4** could explain its moderate affinity to the SIRT2 active site. Moreover, the orientation of the dimethylamino group attached to the pyridine moiety of **STC4** was not in the correct direction toward the selectivity pocket. On the other hand, extension of the dimethylamino group to diethylamino could cause it to interfere with the amino acid Leu138, which may decrease affinity to the SIRT2 active site, as seen for **STC9**. Furthermore, binding free energies of the top-ranked docking positions for re-docked **SirReal2**, **TH-3**, and **STC4** were −9.97, −8.86, and −7.96 kcal/mol, respectively. These binding affinities correlated well with the in vitro SIRT2 inhibition assay (Table 1), where **TH-3** inhibited the SIRT2 enzyme more strongly than **STC4**.

Figure 4D,E illustrated the detailed binding of **STC4** to the SIRT2 active site. The ester moiety formed two hydrogen bond interactions with amino acid residue Asn168. These interactions may explain the higher activity of **STC11**, where the hydrazide moiety in **STC11** can form stronger hydrogen bonds with amino acid Asn168 than the ester moiety of **STC4**. In addition, the trityl moiety formed a CH-π interaction with a key amino acid residue (His178) in the acetyl-lysine channel and blocked the substrate binding site, similar to that described previously for **TH-3**. These binding interactions may be responsible for the SIRT2 inhibition potency of **STC4**. However, the absence of the interactions between the dimethylamino group of **STC4** and key amino acid residues (Tyr139 and Pro140) in the selectivity pocket may explain the 10-fold decrease in SIRT2 inhibition activity of **STC4** compared with that of **TH-3**. Moreover, the lower affinities of both **STC4** and **TH-3** compared with **SirReal2** partially were due to their inability to form a hydrogen bond with the conserved water molecule (W540). Our in silico analysis elucidated the potential SIRT2 inhibitory mechanism of **STC4** mainly through acetyl substrate competition. Despite further structural modifications required to enhance **STC4** potency, these results demonstrated that **STC4** is a promising SIRT2 inhibitor.

Finally, we used ADMET predict 9 software to account for the solubility of **STC4** and its parent compound. The main parameter *S+Sw* that expresses native water solubility, regardless of pH value, was −0.982 and −1.956 for **STLC** and **STC4**, respectively. The optimal value ranged from −2.406 to −0.982, which meant that **STLC** was on the borderline limit; however, **STC4** was in the middle of the optimal range. To confirm that the **STLC** solubility profile is enhanced by introducing the aforementioned derivatizations, we compared solubility profiles of **STLC** and **STC4**. Herein, we showed the solubility change across a wide range of pH values (Figure 5). **STLC** had limited solubility over a wide pH range of 2–8. The predicted pKa values of the acidic carboxyl and basic amino groups appear as dotted red and blue lines, respectively. As for **STC4**, the solubility was remarkably enhanced over a wider pH range (0–7). The predicted pKa values of the three basic amino groups are marked by blue dotted lines. In general, modification of the zwitterion radicals of **STLC** potentially improved solubility and, hence, the pharmacodynamics features.

## 3. Materials and Methods

### 3.1. Chemistry

All reactions were performed in an efficient fume hood. Chemicals were purchased from Sigma-Aldrich (St. Louis, MO, USA), Fluka (Buchs, Switzerland), Kanto Chemical (Tokyo, Japan), Nacalai Tesque (Kyoto, Japan), Tokyo Chemical Industry (Tokyo, Japan), and Wako (Osaka, Japan). Commercially available reagent-grade chemicals were used without further purification. Reaction progress was monitored by thin-layer chromatography (TLC) on precoated plates (Merck, St. Louis, MO, USA), TLC 60 F254 silica sheets, and Fuji Silysia Chemical (Kasugai, Japan) TLC Chromatorex NH silica sheets. Flash column chromatography was carried out on Silica Gel 60N (40–100 mesh, Kanto Chemical, Tokyo, Japan)or NH silica gel Chromatorex (NH, 100–200 mesh, Fuji Silysia Chemical, Kasugai, Japan). Melting points were determined on a melting point apparatus (Yanaco, Kyoto, Japan) and were uncorrected. ^1^H- and ^13^C-NMR spectra were obtained using a Bruker Avance 600 (Billerica, MA, USA) (600 MHz). Chemical shifts were referenced totetramethylsilane. Mass spectra (MS) and high-resolution mass spectra (HRMS) were recorded on a JEOL JMS-DX303HF (Tokyo, Japan) using positive fast atom bombardment (FAB) with 3-nitrobenzyl alcohol as the matrix. Spectral charts are available at the Appendix A.

#### Synthetic Procedures

S-Trityl-l-Cysteine Methyl Ester Hydrochloride **(1)**. This compound was synthesized and obtained as a white solid in a quantitative yield as described previously by Swarbrick et al. [37]. Spectral data of (**1**) were in accordance with those reported in literature (Appendix A).

General procedures for reductive alkylation of compound **(1)** (synthesis of **STC1–9**) [38] were as follows. Compound (**1**) (94.25 mg, 0.25 mmol) and the relevant aldehyde (1.1 equiv.) were dissolved in dry 1,2-DCE and then treated with solid sodium triacetoxyborohydride (74.16 mg, 1.4 equiv.). The reaction mixture was stirred at room temperature, and the reaction progress was monitored by TLC (using NH silica plates; eluent Hex/EA 2:1). Most of the reactions were completed in 3 h. The reaction was then quenched with saturated sodium hydrogen carbonate solution, which was then extracted three times with DCM. The combined organic layers were dried over anhydrous sodium sulfate, and the solvent was removed under reduced pressure. Each product was purified by flash chromatography (NH silica) using the appropriate solvent system.

Methyl *N*-((6-bromopyridin-2-yl)methyl)-*S*-trityl-l-cysteinate (**STC1**). **STC1** was prepared using 6-bromopicolinaldehyde; elution with Hex/EA 7:3 afforded **2a** as a colorless oil (54.61 mg, 40%). ^1^H-NMR (600 MHz, CDCl_3_) *δ* 2.53 (qd, *J* = 12.6, 6.1 Hz, 2H)), 3.07 (dd, *J* = 7.0, 6.1 Hz, 1H), 3.65 (s, 3H), 3.67 (d, *J* = 15.1 Hz, 1H), 3.81 (d, *J* = 15.1 Hz, 1H), 7.18–7.22 (m, 3H), 7.24–7.28 (m, 7H), 7.33 (dd, *J* = 7.7, 1.9 Hz, 2H), 7.39–7.43 (m, 6H), 7.48 (t, *J* = 7.7 Hz, 1H). ^13^C-NMR (150 MHz, CDCl_3_) δ 34.70, 52.08, 52.58, 60.19, 66.87, 120.86, 126.32, 126.75, 127.95, 129.62, 138.81, 141.44, 144.57, 161.08, 173.25. FAB-MS (m/z) 569 (M + Na)^+^; HRFAB-MS calculated for C_29_H_27_BrN_2_O_2_SNa: 569.0874. Found: 569.0878. (Appendix A).

Methyl *N*-(4-(diethylamino)benzyl)-*S*-trityl-L-cysteinate (**STC2**). **STC2** was prepared using 4-(diethylamino)benzaldehyde; elution with Hex/EA 7:3 afforded **2b** as a colorless oil (26.91 mg, 20%). ^1^H-NMR (600 MHz, CDCl_3_) *δ* 1.13 (t, *J* = 7.1 Hz, 6H), 2.47 (dd, *J* = 6.5, 2.6 Hz, 2H), 3.13 (t, *J* = 6.5 Hz, 1H), 3.32 (q, *J* = 7.1 Hz, 4H), 3.44 (d, *J* = 12.6 Hz, 1H), 3.52 (d, *J* = 12.6 Hz, 1H), 3.65 (s, 3H), 6.60 (d, *J* = 8.7 Hz, 2H), 7.07 (d, *J* = 8.7 Hz, 2H), 7.19–7.22 (m, 3H), 7.24–7.28 (m, 6H), 7.38–7.44 (m, 6H). ^13^C-NMR (150 MHz, CDCl_3_) δ 12.59, 34.69, 44.43, 51.28, 51.86, 59.57, 66.67, 111.91, 126.66, 127.90, 129.44, 129.66, 144.67, 147.06, 173.91. FAB-MS (m/z) 561 (M + Na)^+^; HRFAB-MS calculated for C_34_H_38_N_2_O_2_SNa: 561.2552. Found: 561.2566. (Appendix A).

Methyl *N*-(4-(dimethylamino)benzyl)-*S*-trityl-l-cysteinate (**STC3**). **STC3** was prepared using 4-(dimethylamino)benzaldehyde; elution with Hex/EA 7:3 afforded **2c** as a colorless oil (31.87 mg, 25%). ^1^H-NMR (600 MHz, CDCl_3_) *δ* 2.47 (dd, *J* = 6.5, 3.0 Hz, 2H), 2.91 (s, 6H), 3.13 (t, *J* = 6.5 Hz, 1H), 3.47 (d, *J* = 12.6 Hz, 1H), 3.55 (d, *J* = 12.6 Hz, 1H), 3.65 (s, 3H), 6.67 (d, *J* = 8.7 Hz, 2H), 7.11 (d, *J* = 8.7 Hz, 2H), 7.20 (ddd, *J* = 7.3, 3.9, 1.2 Hz, 3H), 7.24–7.28 (m, 6H), 7.38–7.42 (m, 6H). ^13^C-NMR (150 MHz, CDCl_3_) δ 34.71, 40.80, 51.27, 51.88, 59.57, 66.69, 112.72, 126.67, 127.91, 129.21, 129.66, 144.68, 149.98, 173.88. FAB-MS (m/z) 533 (M + Na)^+^; HRFAB-MS calculated for C_32_H_34_N_2_O_2_SNa: 533.2239. Found: 533.2557. (Appendix A).

Methyl *N*-((4-(dimethylamino)pyridin-2-yl)methyl)-*S*-trityl-l-cysteinate (**STC4**). **STC4** was prepared using 4-(dimethylamino)picolinaldehyde; elution with Hex/EA 1:1 afforded **2d** as a colorless oil (44.71, 35%). ^1^H-NMR (600 MHz, CDCl_3_) *δ* 2.49–2.55 (m, 1H), 3.14 (t, *J* = 6.7 Hz, 1H), 2.93 (s, 6H), 3.58 (d, *J* = 14.4 Hz, 1H), 3.64 (s, 3H), 3.75 (d, *J* = 14.4 Hz, 1H), 6.35 (dd, *J* = 6.0, 2.6 Hz, 1H), 6.66 (d, *J* = 2.6 Hz, 1H), 7.18–7.20 (m, 3H), 7.24–7.26 (m, 6H), 7.38–7.40 (m, 6H), 8.12 (d, *J* = 6.0 Hz, 1H). ^13^C-NMR (150 MHz, CDCl_3_) *δ* 34.91, 39.09, 51.92, 53.46, 60.21, 66.78, 104.71, 105.15, 126.68, 127.89, 129.66, 144.67, 149.24, 154.99, 159.14, 173.67. FAB-MS (m/z) 512.5 (M + H)^+^; HRFAB-MS calculated for C_31_H_33_N_3_O_2_S: 512.2372. Found: 512.2385. (Appendix A).

Methyl *N*-(2-chloro-4-(dimethylamino)benzyl)-*S*-trityl-l-cysteinate (**STC5**). **STC5** was prepared using 2-chloro-4-(dimethylamino)benzaldehyde; elution with Hex/EA 7:3 afforded **2d** as a colorless oil (34.10, 25%). ^1^H-NMR (600 MHz, CDCl_3_) *δ* 2.46–2.52 (m, 2H), 2.91 (s, 6H), 3.08 (t, *J* = 6.5 Hz, 1H), 3.59–3.67 (m, 2H), 3.64 (s, 3H), 6.53 (dd, *J* = 8.5, 2.6 Hz, 2H), 6.65 (d, *J* = 2.6 Hz, 1H), 7.10 (d, *J* = 8.5 Hz, 1H), 7.17–7.23 (m, 3H), 7.25–7.28 (m, 6H), 7.40–7.42 (m, 6H). ^13^C-NMR (150 MHz, CDCl_3_) *δ* 31.60, 34.69, 40.45, 48.81, 51.90, 59.71, 66.71, 110.94, 112.98, 124.07, 126.67, 127.91, 129.66, 130.95, 134.65, 144.68, 150.68, 173.63. FAB-MS (m/z) 567.5 (M + Na)^+^; HRFAB-MS calculated for C_32_H_33_ClN_2_O_2_SNa: 567.1849. Found: 567.1859. (Appendix A).

Methyl *N*-(2-methoxy-4-(dimethylamino)benzyl)-*S*-trityl-l-cysteinate (**STC6**). **STC6** was prepared using 2-methoxy-4-(dimethylamino)benzaldehyde; elution with Hex/EA 2:1 afforded **2d** as a colorless oil (29.73 mg, 22%). ^1^H-NMR (600 MHz, CDCl_3_) *δ* 2.51 (dd, *J* = 6.6, 1.1 Hz, 2H), 2.96 (s, 6H), 3.13 (t, *J* = 6.6 Hz, 1H), 3.55–3.62 (m, 2H), 3.63 (s, 3H), 3.80 (s, 3H), 6.26 (dd, *J* = 8.2, 2.4 Hz, 1H), 6.24 (d, *J* = 2.4 Hz, 1H), 7.00 (d, *J* = 8.2 Hz, 1H), 7.19–7.25 (m, 3H), 7.28–7.31 (m, 6H), 7.43–7.45 (m, 6H). ^13^C-NMR (150 MHz, CDCl_3_) *δ* 34.68, 40.85, 46.94, 51.79, 55.10, 59.63, 66.61, 96.15, 104.48, 115.89, 126.63, 127.88, 127.94, 129.63, 129.66, 129.66, 129.67, 129.69, 129.70, 129.70, 130.68, 144.75, 151.57, 158.59, 173.85. FAB-MS (m/z) 563.5 (M + Na)^+^; HRFAB-MS calculated for C_33_H_36_N_2_O_3_SNa: 563.2344. Found: 563.2360. (Appendix A).

Methyl *N*-(quinolin-3-ylmethyl)-*S*-trityl-l-cysteinate (**STC7**). **STC7** was prepared using quinoline-3-carbaldehyde; elution with Hex/EA 3:1 afforded **2g** as yellow oil (25.92 mg, 20%). ^1^H-NMR (600 MHz, CDCl_3_) *δ* 2.54 (dd, *J* = 6.5, 2.8 Hz, 1H), 3.07 (t, *J* = 6.5 Hz, 1H), 3.67 (s, 1H), 3.72 (d, *J* = 13.8 Hz, 1H), 3.90 (d, *J* = 13.8 Hz, 1H), 7.16–7.19 (m, 3H), 7.22–7.26 (m, 6H), 8.81 (s, 1H), 7.38–7.42 (m, 6H), 7.53 (ddd, *J* = 8.1, 6.9, 1.2 Hz, 1H), 7.69 (ddd, *J* = 8.4, 6.9, 1.2 Hz, 1H), 7.76 (d, *J* = 8.1 Hz, 1H), 8.07 (d, *J* = 1.2 Hz, 1H), 8.09 (d, *J* = 8.4 Hz, 1H). ^13^C-NMR (150 MHz, CDCl_3_) *δ* 34.79, 49.27, 52.06, 59.73, 66.90, 126.69, 126.74, 126.80, 127.65, 127.93, 129.11, 129.22, 129.60, 132.17, 134.73, 144.53, 147.56, 151.47, 173.57. FAB-MS (m/z) 519.4 (M + H)^+^; HRFAB-MS calculated for C_33_H_31_N_2_O_2_S: 519.2106. Found: 519.2118. (Appendix A).

Methyl *N*-(pyridin-2-ylmethyl)-*S*-trityl-l-cysteinate (**STC8**). **STC8** was prepared using picolinaldehyde; elution with Hex/EA 2:1 afforded **2g** as yellow oil (37.44 mg, 32%) ^1^H-NMR (600 MHz, CDCl_3_) *δ* 2.44–2.50 (m, 2H), 3.05 (t, *J* = 6.6 Hz, 1H), 3.57 (d, *J* = 0.4 Hz, 3H), 3.62 (d, *J* = 14.3 Hz, 1H), 3.73 (d, *J* = 14.3 Hz, 1H), 7.04–7.06 (m, 1H), 7.11–7.13 (m, 3H), 7.17–7.19 (m, 6H), 7.23 (d, *J* = 7.6 Hz, 1H), 7.33–7.35 (m, 6H), 7.53 (td, *J* = 7.6, 1.5 Hz, 1H), 8.43–8.44 (m, 1H). ^13^C-NMR (150 MHz, CDCl_3_) *δ* 34.68, 51.98, 53.15, 60.29, 66.80, 122.00, 122.18, 126.71, 127.93, 129.65, 136.45, 144.63, 149.16, 159.14, 173.47. FAB-MS (m/z) 569.1 (M + H)^+^; HRFAB-MS calculated for C_29_H_29_N_2_O_2_S: 469.1950. Found: 469.1956. (Appendix A).

Methyl *N*-((4-(diethylamino)pyridin-2-yl)methyl)-*S*-trityl-l-cysteinate (**STC9**). **STC9** was prepared using 4-(diethylamino)picolinaldehyde; elution with Hex/EA 1:1 afforded **2d** as a colorless oil (41.77 mg, 31%). ^1^H-NMR (600 MHz, CDCl_3_) *δ* 8.08 (d, *J* = 6.0 Hz, 1H), 7.38–7.40 (m, 6H), 7.24–7.27 (m, 6H), 7.18–7.21 (m, 3H), 6.61 (d, *J* = 2.6 Hz, 1H), 6.32 (dd, *J* = 6.0, 2.6 Hz, 1H), 3.73 (d, *J* = 14.3 Hz, 1H), 3.63 (s, 3H), 3.55 (d, *J* = 14.3 Hz, 1H), 3.35 (dq, *J* = 14.3, 7.1 Hz, 1H), 3.26 (dq, *J* = 14.3, 7.1 Hz, 1H), 3.13 (t, *J* = 6.7 Hz, 1H), 2.51 (dd, *J* = 6.7, 2.6 Hz, 1H), 1.12 (t, *J* = 7.1 Hz, 1H). ^13^C-NMR (150 MHz, CDCl_3_) *δ* 12.38, 34.88, 43.72, 51.91, 53.44, 60.13, 66.77, 104.27, 104.80, 127.88, 129.66, 144.66, 152.76, 159.20, 173.66. FAB-MS (m/z) 540.2 (M + H)^+^; HRFAB-MS calculated for C_33_H_38_N_3_O_2_S: 540.2685. Found: 540.2720. (Appendix A).

3-(Dimethylamino)picolinic acid (**2**). A solution of 4-chloropicolinic acid (200 mg, 1.25 mmol) in aqueous dimethylamine (40%, 3.1 mL) was stirred at 150 °C for 2 h in a sealed tube. The mixture was concentrated in vacuo, dissolved in EtOAc (20 mL), and washed with saturated aqueous NaHCO_3_ (20 mL). The organic phase was dried over Na_2_SO_4_ and evaporated in vacuo to afford **2** (190 mg, 90%) as a white solid [34]. ^1^H-NMR (600 MHz, DMSO) *δ* 3.08 (s, 6H), 6.77 (brs, 1H), 7.29 (brs, 1H), 8.09 (brs, 1H). m.p. > 300 °C. FAB-MS (m/z) 167.0 (M + H)^+^. (Appendix A).

3-(Dimethylamino)picolinoyl chloride (**3**). Compound **2** (190 mg, 1.1 mmol) was refluxed with 5 mL of SOCl_2_ in the presence of a catalytic amount of dimethylformamide for 10 h. The mixture was concentrated and dried to afford a yellow solid **(3)** in a quantitative yield, which was used without further purification [31].

Methyl *N*-(4-(dimethylamino)picolinoyl)-*S*-trityl-l-cysteinate (**STC10**) [35]. A batch of compound (**1**) (94.25 mg, 0.25 mmol) and TEA (1.5 equiv.) were mixed in DCM at 0 °C. Then, the freshly synthesized acyl chloride **(3)** (46.15 mg, 0.25 mmol) in DCM was added dropwise. The mixture was stirred overnight at room temperature then analyzed by TLC. Upon reaction completion, the crude mixture was washed with water, and the organic layer was dried over anhydrous sodium sulfate. The residue was applied to an NH silica column chromatography to obtain a pure product using Hex/EA 1:1 eluent to afford **STC10** as a white solid (35.45 mg, 27%). ^1^H-NMR (600 MHz, CDCl_3_) *δ* 2.68–2.76 (m, 2H), 3.05 (s, 6H), 3.71 (s, 3H), 4.69–4.72 (m, 1H), 6.56 (dd, *J* = 8.1, 2.8 Hz, 1H), 7.17–7.20 (m, 3H), 7.23–7.26 (m, 6H), 7.39–7.41 (m, 6H), 8.20 (d, *J* = 2.8 Hz, 1H), 8.63 (d, *J* = 8.1 Hz, 1H). ^13^C-NMR (150 MHz, CDCl_3_) *δ* 34.19, 39.27, 51.44, 52.53, 66.99, 105.36, 108.29, 126.79, 127.96, 129.62, 144.43, 148.49, 149.46, 155.20, 165.05, 170.84. m.p. 54–55 °C. FAB-MS (m/z) 526.2 (M + H)^+^; HRFAB-MS calculated for C_31_H_32_N_3_O_3_S: 526.2164. Found: 526.2168. (Appendix A).

*(S)-*2-(((4-(dimethylamino)pyridin-2-yl)methyl)amino)-3-(tritylthio)propanehydrazide (**STC11**). **STC11** was prepared by reacting **STC4** (30.20 mg, 0.05 mmol) with hydrazine hydrate in EtOH, in accordance with previously reported procedures [39]. A white solid product was obtained in a quantitative yield. ^1^H-NMR (600 MHz, CDCl_3_) *δ* 2.23 (brs, 2H), 2.56 (dd, *J* = 12.8, 8.8 Hz, 1H), 2.70 (dd, *J* = 12.8, 4.3 Hz, 1H), 2.70 (dd, *J* = 12.8, 4.3 Hz, 1H), 2.88 (dd, *J* = 8.8, 4.3 Hz, 1H), 2.97 (s, 6H), 3.53–3.59 (m, 2H), 6.38 (dd, *J* = 6.0, 2.6 Hz, 1H), 6.42 (d, *J* = 2.6 Hz, 1H), 7.18–7.21 (m, 3H), 7.25–7.28 (m, 6H), 7.40–7.42 (m, 6H), 8.16 (d, *J* = 6.0 Hz, 1H), 8.65 (s, 1H). ^13^C-NMR (150 MHz, CDCl_3_) *δ* 35.06, 39.17, 53.56, 60.28, 66.99, 104.79, 105.29, 126.79, 127.98, 129.65, 144.56, 148.87, 155.06, 158.38, 172.61. m.p. 162–164 °C. FAB-MS (m/z) 534.3 (M+Na)^+^; HRFAB-MS calculated for C_30_H_33_N_5_OSNa: 534.2304. Found: 534.2301. (Appendix A).

### 3.2. Biological Assays

#### 3.2.1. In Vitro Inhibitory Activities against SIRT2

Deacetylase activities of SIRT2 were measured with an electrophoretic mobility shift assay [24,40]. For the electrophoretic mobility shift assay to measure SIRT2 activity, recombinant SIRT2 proteins were incubated with a carboxyfluorescein (FAM)-labeled fluorescent peptide (FAM-RHKK(Ac)LM) and 1 mM NAD in 50 μL assay buffer (25 mM Tris-HCl, (pH 9.0), 137 mM NaCl, 2.7 mM KCl, 1 mM MgCl_2_, 0.1 mg/mL bovine serum albumin) in 384-well plates. After 60 min at 37 °C, the reaction was stopped by adding nicotinamide (final concentration 10 mM) in 50 μL stop buffer (100 mM HEPES (pH 7.5), 10 mM EDTA, 0.25% CR-3). The samples were analyzed using a LabChip EZ Reader II (PerkinElmer, Waltham, MA, USA). Percent conversion was defined as 100 × P / (P + S), where P and S are peak heights of the product and peptide substrate, respectively. The IC_50_ values were determined as means with standard deviation calculated from at least three independent dose–response curves using Origin software (OriginLab, Northampton, MA, USA).

#### 3.2.2. Cell Culture and Drug Treatment

HeLa human cervical carcinoma cell lines were incubated in Dulbecco’s modified Eagle’s medium. MCF-7 human breast cancer, leukemic (K562, HL-60, MT-2, and Jurkat), and normal blood (PBMC) (Precision Bioservices, Frederic, MD, USA) cells were incubated in RPMI1640 medium. All media (Wako, Osaka, Japan) were supplemented with 10% fetal bovine serum (Sigma-Aldrich, St. Louis, MO, USA) and 89 μg/mL streptomycin (Meiji Seika, Tokyo, Japan) at 37 °C in a humidified atmosphere of 95% air and 5% CO_2_. Exponentially growing cells were cultured in 24-well and 96-well plates (Iwaki brand Asahi Glass, Tokyo, Japan) at 2 × 10^4^ cells/mL and 1 × 10^6^ cells/mL, respectively, for 96 h before the addition of drugs (optimum cell number for cytotoxicity assays was determined in preliminary experiments). Stock solutions of compounds in concentrations between 0.01–10 mM were prepared in dimethyl sulfoxide (DMSO; Wako, Osaka, Japan) and further dilution was made with fresh culture medium. The concentration of DMSO in the final culture medium was 1%, which had no effect on cell viability [41].

#### 3.2.3. MTT Assay

The level of cellular MTT (Dojindo, Kumamoto, Japan) reduction was quantified as described previously [42,43] After 24 h of preincubation at 37 °C, cells were exposed to various concentrations (0.1–100 μM) of the tested compounds and SirReal2 (positive control) for 96 h. At the end of this period, cells were stained with MTT solution and incubated for an additional 4 h at 37 °C. After the medium was removed, the formazan crystals were solubilized by addition of 100 μL DMSO to each well, and absorbance was determined using an Infinitive M1000 plate reader (Tecan, Mannedorf, Switzerland) at a wavelength of 550 nm with background subtraction at 630 nm. Every concentration was repeated in three wells, and IC_50_ values were calculated from MTT results and defined as the drug concentrations that reduced absorbance to 50% of the control values. 

#### 3.2.4. DNA Cleavage

The DNA cleavage assays were performed as described previously [32,44,45]. Intensity of the bands was quantitated using ImageJ software (NIH, Bethesda, MD, USA).

### 3.3. Molecular Docking

Sirt2 X-ray structure Sirt2-SirReal2-NAD (PDB code 4RMG) [30] was used in the present study. The protein structure was prepared using the structure preparation module in MOE (Version 2019.01, Chemical Computing Group Inc., Montreal, QC, Canada). Water molecules and ligand atoms, except the zinc ion and the conserved water molecule bridging the interaction between Pro94 and the carbonyl group of **SirReal2,** were removed from the structure. Docking studies were performed using the rigid-receptor method [46,47]. The co-crystallized ligand (**SirReal2**) was defined as the center of the binding site. Three hundred docking positions were generated for each ligand. All other options were left at their default values [48]. The co-crystallized ligand was also docked with other compounds for validation of the docking method. The binding free energy (ΔG) in kcal/mol of the re-docked **SirReal2** and the inhibitors of this study were calculated using the top-scoring docking positions. The generated docking positions were visualized using MOE [49].

## 4. Conclusions

In this study, we found a new target for **STLC**-derived compounds distinct from their well-known mitotic kinesin Eg5 inhibition. The introduced derivatizations not only conferred **STLC** with SIRT2 inhibitory effect but also addressed its pharmacokinetic issue and enhanced its water solubility. Three compounds showed pronounced SIRT2 inhibition and antiproliferative effects against different cancer cell lines, making them promising candidates for further optimization.

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
