# Peer review of "Antiproliferative S-Trityl-l-Cysteine -Derived Compounds as SIRT2 Inhibitors: Repurposing and Solubility Enhancement"

_molecules, 2019, doi:10.3390/molecules24183295_

Round 1

Reviewer 1 Report

In the manuscript (ID:molecules-572679), authors Radwan et al describe their efforts in developing a novel class of SIRT2 inhibitors by combining chemical properties of the previously identified Eg5 inhibitor STLC, and their previously described potent inhibitor class based on a trityl-histidine scaffold. In the process, the authors claim to have improved the solubility issues of STLC, and also imparted a novel bioactivity on their novel compounds.

Overall, the work has merit, and their newly identified compounds would seem to deserve further exploration as promising candidates.

In my opinion, there are a few issues that need to be addressed, before this manuscript can be accepted for publication in this journal.

1) It wasn’t clear to me, either from the title, or abstract, what the stated purpose or goal of this work was; was it to obtain better SIRT2 inhibitors, by incorporating properties of STLC, or enhance anti-cancer and cytotoxic properties of STLC and also address its solubility issues by modifying its primary amino group based on their TH-3 compounds? I ask this because at one place in the abstract, the authors state that their “derivatives conferred a new and unexpected SIRT2 bioactivity on STLC”. Why was this so unexpected?

2) In many places, the language seems colloquial and the syntax is confusing. This may be rectified by a thorough proof-reading exercise, throughout the manuscript. A few examples are noted below:

Abstract: “On the other side; such a Zwitterion structure….” Should be “On the other hand, such…..”

Introduction: “Recently, we have disclosed a new class….” Should be “Recently, we have identified and characterized…..”

Last sentence of Intro: “Thus our work aims at conferring STLC a new and unexpected bioactivity against SIRT2” This is highly confusing. If their work actually aimed at conferring anti-SIRT2 bioactivity, then why call it new and “unexpected”?

Conclusion: The first sentence “We have found an unexpected……….away from mitotic kinesis Eg5” should be “We have found……….distinct from mitotic kinesin Eg5”

Last sentence: “………making them considerable hits for further optimization” should be “making them promising hits/candidates for further optimization”.

 3) Figure 3 (DNA cleavage assay) – I assume that the bands marked as Form I represent the uncleaved, supercoiled plasmid. However, I could not distinguish between Form II and Form III, especially since there is no molecular weight reference, or any description in the text or figure legend. Secondly, it is claimed that cleavage activity by STC9 is strongest, above STC4 and STC11. That is quite evident from visual inspection of band intensities. It would help if there was some quantification of the band intensities was provided to justify the claim.

4) Figure 5: The SP profiles are not labeled, so it is left for the readers to guess which profile represents STLC and STC4 respectively. They can be labeled as panels A and B, clearly stating in the figure legend which profile belongs to which compound.

Author Response

Comment 1. It wasn’t clear to me, either from the title, or abstract, what the stated purpose or goal of this work was; was it to obtain better SIRT2 inhibitors, by incorporating properties of STLC, or enhance anti-cancer and cytotoxic properties of STLC and also address its solubility issues by modifying its primary amino group based on their TH-3 compounds? I ask this because at one place in the abstract, the authors state that their “derivatives conferred a new and unexpected SIRT2 bioactivity on STLC”. Why was this so unexpected?

The goal of this work is to have a new class of drug-like Sirt2 inhibitors based on STLC scaffold. To clarify this goal, the words “new and unexpected” were omitted in the sentence pointed out. Furthermore, new sentence “Therefore, we could obtain drug-like STLC derivatives that work by a new mechanism of action.” was added in the later part of abstract (p2, lane 15 from the top).

Comment 2. In many places, the language seems colloquial and the syntax is confusing. This may be rectified by a thorough proof-reading exercise, throughout the manuscript. A few examples are noted below:

Abstract: “On the other side; such a Zwitterion structure….” Should be “On the other hand, such…..”

Introduction: “Recently, we have disclosed a new class….” Should be “Recently, we have identified and characterized…..”

Last sentence of Intro: “Thus our work aims at conferring STLC a new and unexpected bioactivity against SIRT2” This is highly confusing. If their work actually aimed at conferring anti-SIRT2 bioactivity, then why call it new and “unexpected”?

Conclusion: The first sentence “We have found an unexpected……….away from mitotic kinesis Eg5” should be “We have found……….distinct from mitotic kinesin Eg5”

Last sentence: “………making them considerable hits for further optimization” should be “making them promising hits/candidates for further optimization”.

Unnatural English expression, including sentences pointed out, was rectified throughout the manuscript. This was performed by the native English writer in Medical English Service, Kyoto, Japan.

Comment 3. Figure 3 (DNA cleavage assay) – I assume that the bands marked as Form I represent the uncleaved, supercoiled plasmid. However, I could not distinguish between Form II and Form III, especially since there is no molecular weight reference, or any description in the text or figure legend. Secondly, it is claimed that cleavage activity by STC9 is strongest, above STC4 and STC11. That is quite evident from visual inspection of band intensities. It would help if there was some quantification of the band intensities was provided to justify the claim.

About distinction between Form II and Form III, explanation was added in the legend for Figure 3A. Intensity of bands was quantitated, and ratio of Form I and Form II was newly summarized in the graph (Figure 3B).

Comment 4. Figure 5: The SP profiles are not labeled, so it is left for the readers to guess which profile represents STLC and STC4 respectively. They can be labeled as panels A and B, clearly stating in the figure legend which profile belongs to which compound.

Labels (A) and (B) were added in Figure 5 to clarify which profiles belongs to which compound.

Reviewer 2 Report

The study appears to be reasonably well executed and presented; some additions/alterations to the docking part would be worthwhile:

Please verify that the docking method employed can reproduce the experimental binding mode of SirReal2, with and without the conserved water mentioned below In addition to the interaction analysis performed, please provide docking scores for the molecules investigated, and comment on whether these relate with in vitro IC50s, which in turn, could provide opportunity to incorporate this docking approach into the rational design and selection of further inhibitors In the Methods it is noted that "the conserved water molecule bridging
the interaction between Pro94 and the carbonyl group of SirReal2" was retained for docking, however, it does not appear that the docked ligands interact with this; is there any difference in the docking result when it is performed without this water present? It would be worth repeating the docking - including validation of docking with SirReal2 - without this water present

Author Response

Comment 1. Please verify that the docking method employed can reproduce the experimental binding mode of SirReal2, with and without the conserved water mentioned below.

This point was verified and the following new sentences were added to the manuscript with citing new paper (Ref 36).

“The co-crystallized ligand SirReal2 was firstly re-docked in its corresponding co-crystal structure (PDB code 4RMG) to test whether the Molecular Operating Environment (MOE) is able to accurately reproduce the correct binding mode of the inhibitor. We kept the conserved structural bridging water molecule (W540) due to its critical importance for the ligand binding affinity to SIRT2 active site as experimentally confirmed in a previous SAR study of potent SIRT2 inhibitors. Figure 4A shows a superimposition of the co-crystallized ligand SirReal2 and its superposed docking conformation, where SirReal2 is perfectly docked into its crystal structure with a root-mean-squared deviation value 0.16 Å and formed a hydrogen bond with the conserved water molecule (W540) similar to the co-crystallized ligand.” (p7, lane 2 from the bottom-p8, lane 7 from the top)

“Moreover, the lower affinities of both STC4 and TH-3 compared SirReal2 is partially due to their inability to form hydrogen bonding with the conserved water molecule (W540).” (p8, lane 4-6 from the bottom)

“The co-crystallized ligand was also docked with other compounds for validation of the docking method. The binding free energy (DG) in kcal/mol of the re-docked SirReal2 and the inhibitors were calculated using the top-scored docking posisions.” (p15, lane 6-9 from the top)

Furthermore, Figure 4A was added for more accuracy.

Comment 2. In addition to the interaction analysis performed, please provide docking scores for the molecules investigated, and comment on whether these relate with in vitro IC50s, which in turn, could provide opportunity to incorporate this docking approach into the rational design and selection of further inhibitors.

The following new sentences were added to the manuscript.

“Furthermore, binding free energies of the top-ranked docking poses for re-docked SirReal2, TH-3 and STC4 were – 9.97, – 8.86, and – 7.96 Kcal/mol, respectively. These binding affinities correlated well with the in vitro SIRT2 inhibition assay (Table 1), where TH-3 inhibits SIRT2 enzyme stronger than STC4.” (p8, lane 17-20 from the top)

“The binding free energy (DG) in kcal/mol of the re-docked SirReal2 and the inhibitors of this study were calculated using the top-scored docking positions.” (p15, lane 7-9 from the top)

Comment 3. In the Methods it is noted that "the conserved water molecule bridging the interaction between Pro94 and the carbonyl group of SirReal2" was retained for docking, however, it does not appear that the docked ligands interact with this.

This point was verified and the following new sentences were added to the manuscript.

“Moreover, the lower affinities of both STC4 and TH-3 compared SirReal2 is partially due to their inability to form hydrogen bonding with the conserved water molecule (W540).” (p8, lane 4-6 from the bottom)

Also, figure 4D was modified by showing the conserved water molecule (W540):

Comment 4. Is there any difference in the docking result when it is performed without this water present?

See the response to comment 1.

Comment 5. It would be worth repeating the docking - including validation of docking with SirReal2 - without this water present.

See the response to comment 1.